# On Iterative Hard Thresholding Methods for High-dimensional M-Estimation

**Prateek Jain**[*]  **Ambuj Tewari**[†]  **Purushottam Kar**[*]

[*]Microsoft Research, INDIA
[†]University of Michigan, Ann Arbor, USA
{prajain,t-purkar}@microsoft.com, tewaria@umich.edu

## Abstract

The use of M-estimators in generalized linear regression models in high dimensional settings requires risk minimization with hard $L_0$ constraints. Of the known methods, the class of projected gradient descent (also known as iterative hard thresholding (IHT)) methods is known to offer the fastest and most scalable solutions. However, the current state-of-the-art is only able to analyze these methods in extremely restrictive settings which do not hold in high dimensional statistical models. In this work we bridge this gap by providing the first analysis for IHT-style methods in the high dimensional statistical setting. Our bounds are tight and match known minimax lower bounds. Our results rely on a general analysis framework that enables us to analyze several popular hard thresholding style algorithms (such as HTP, CoSaMP, SP) in the high dimensional regression setting. Finally, we extend our analysis to the problem of low-rank matrix recovery.

## 1  Introduction

Modern statistical estimation is routinely faced with real world problems where the number of parameters $p$ handily outnumbers the number of observations $n$. In general, consistent estimation of parameters is not possible in such a situation. Consequently, a rich line of work has focused on models that satisfy special structural assumptions such as sparsity or low-rank structure. Under these assumptions, several works (for example, see [1, 2, 3, 4, 5]) have established that consistent estimation is information theoretically possible in the "$n \ll p$" regime as well.

The question of efficient estimation, however, is faced with feasibility issues since consistent estimation routines often end-up solving NP-hard problems. Examples include sparse regression which requires loss minimization with sparsity constraints and low-rank regression which requires dealing with rank constraints which are not efficiently solvable in general [6].

Interestingly, recent works have demonstrated that these hardness results can be avoided by assuming certain natural conditions over the loss function being minimized such as restricted strong convexity (RSC) and restricted strong smoothness (RSS). The estimation routines proposed in these works typically make use of convex relaxations [5] or greedy methods [7, 8, 9] which do not suffer from infeasibility issues.

Despite this, certain limitations have precluded widespread use of these techniques. Convex relaxation-based methods typically suffer from slow rates as they solve non-smooth optimization problems apart from being hard to analyze in terms of global guarantees. Greedy methods, on the other hand, are slow in situations with non-negligible sparsity or relatively high rank, owing to their incremental approach of adding/removing individual support elements.

Instead, the methods of choice for practical applications are actually projected gradient (PGD) methods, also referred to as iterative hard thresholding (IHT) methods. These methods directly project

the gradient descent update onto the underlying (non-convex) feasible set. This projection can be performed efficiently for several interesting structures such as sparsity and low rank. However, traditional PGD analyses for convex problems viz. [10] do not apply to these techniques due to the non-convex structure of the problem.

An exception to this is the recent work [11] that demonstrates that PGD with non-convex regularization can offer consistent estimates for certain high-dimensional problems. However, the work in [11] is only able to analyze penalties such as SCAD, MCP and capped $L_1$. Moreover, their framework cannot handle commonly used penalties such as $L_0$ or low-rank constraints.

**Insufficiency of RIP based Guarantees for M-estimation.** As noted above, PGD/IHT-style methods have been very popular in literature for sparse recovery and several algorithms including Iterative Hard Thresholding (IHT) [12] or GraDeS [13], Hard Thresholding Pursuit (HTP) [14], CoSaMP [15], Subspace Pursuit (SP) [16], and OMPR($\ell$) [17] have been proposed. However, the analysis of these algorithms has traditionally been restricted to settings that satisfy the Restricted Isometry property (RIP) or incoherence property. As the discussion below demonstrates, this renders these analyses inaccessible to high-dimensional statistical estimation problems.

All existing results analyzing these methods require the condition number of the loss function, restricted to sparse vectors, to be smaller than a universal constant. The best known such constant is due to the work of [17] that requires a bound on the RIP constant $\delta_{2k} \leq 0.5$ (or equivalently a bound $\frac{1+\delta_{2k}}{1-\delta_{2k}} \leq 3$ on the condition number). In contrast, real-life high dimensional statistical settings, wherein pairs of variables can be arbitrarily correlated, routinely require estimation methods to perform under arbitrarily large condition numbers. In particular if two variates have a covariance matrix like $\begin{bmatrix} 1 & 1-\epsilon \\ 1-\epsilon & 1 \end{bmatrix}$, then the restricted condition number (on a support set of size just 2) of the sample matrix cannot be brought down below $1/\epsilon$ even with infinitely many samples. In particular when $\epsilon < 1/6$, none of the existing results for hard thresholding methods offer *any* guarantees. Moreover, most of these analyses consider only the least squares objective. Although recent attempts have been made to extend this to general differentiable objectives [18, 19], the results continue to require that the restricted condition number be less than a universal constant and remain unsatisfactory in a statistical setting.

**Overview of Results.** Our main contribution in this work is an analysis of PGD/IHT-style methods in statistical settings. Our bounds are tight, achieve known minmax lower bounds [20], and hold for arbitrary differentiable, possibly even *non-convex* functions. Our results hold even when the underlying condition number is arbitrarily large and only require the function to satisfy RSC/RSS conditions. In particular, this reveals that these iterative methods are indeed applicable to statistical settings, a result that escaped all previous works.

Our first result shows that the PGD/IHT methods achieve global convergence if used with a relaxed projection step. More formally, if the optimal parameter is $s^*$-sparse and the problem satisfies RSC and RSS constraints $\alpha$ and $L$ respectively (see Section 2), then PGD methods offer global convergence so long as they employ projection to an $s$-sparse set where $s \geq 4(L/\alpha)^2 s^*$. This gives convergence rates that are identical to those of convex relaxation and greedy methods for the Gaussian sparse linear model. We then move to a family of efficient "fully corrective" methods and show as before, that for arbitrary functions satisfying the RSC/RSS properties, these methods offer global convergence.

Next, we show that these results allow PGD-style methods to offer global convergence in a variety of statistical estimation problems such as sparse linear regression and low rank matrix regression. Our results effortlessly extend to the noisy setting as a corollary and give bounds similar to those of [21] that relies on solving an $L_1$ regularized problem.

Our proofs are able to exploit that even though hard-thresholding is not the prox-operator for any convex prox function, it still provides strong contraction when projection is performed onto sets of sparsity $s \gg s^*$. This crucial observation allows us to provide the first unified analysis for hard thresholding based gradient descent algorithms. Our empirical results confirm our predictions with respect to the recovery properties of IHT-style algorithms on badly-conditioned sparse recovery problems, as well as demonstrate that these methods can be orders of magnitudes faster than their $L_1$ and greedy counterparts.

**Organization.** Section 2 sets the notation and the problem statement. Section 3 introduces the PGD/IHT algorithm that we study and proves that the method guarantees recovery assuming the RSC/RSS property. We also generalize our guarantees to the problem of low-rank matrix regression. Section 4 then provides crisp sample complexity bounds and statistical guarantees for the PGD/IHT estimators. Section 5 extends our analysis to a broad family of compressive sensing algorithms that include the so-called fully-corrective hard thresholding methods and provide similar results for them as well. We present some empirical results in Section 6 and conclude in Section 7.

## 2 Problem Setup and Notations

**High-dimensional Sparse Estimation.** Given data points $X = [X_1, \ldots, X_n]^T$, where $X_i \in \mathbb{R}^p$, and the target $Y = [Y_1, \ldots, Y_n]^T$, where $Y_i \in \mathbb{R}$, the goal is to compute an $s^*$-sparse $\boldsymbol{\theta}^*$ s.t.,

$$\boldsymbol{\theta}^* = \arg \min_{\boldsymbol{\theta}, \|\boldsymbol{\theta}\|_0 \leq s^*} f(\boldsymbol{\theta}). \tag{1}$$

Typically, $f$ can be thought of as an empirical risk function i.e. $f(\boldsymbol{\theta}) = \frac{1}{n} \sum_i \ell(\langle X_i, \boldsymbol{\theta} \rangle, Y_i)$ for some loss function $\ell$ (see examples in Section 4). However, for our analysis of PGD and other algorithms, we need not assume any other property of $f$ other than differentiability and the following two RSC and RSS properties.

**Definition 1** (RSC Property). *A differentiable function $f : \mathbb{R}^p \to \mathbb{R}$ is said to satisfy restricted strong convexity (RSC) at sparsity level $s = s_1 + s_2$ with strong convexity constraint $\alpha_s$ if the following holds for all $\boldsymbol{\theta}_1, \boldsymbol{\theta}_2$ s.t. $\|\boldsymbol{\theta}_1\|_0 \leq s_1$ and $\|\boldsymbol{\theta}_2\|_0 \leq s_2$:*

$$f(\boldsymbol{\theta}_1) - f(\boldsymbol{\theta}_2) \geq \langle \boldsymbol{\theta}_1 - \boldsymbol{\theta}_2, \nabla_{\boldsymbol{\theta}} f(\boldsymbol{\theta}_2) \rangle + \frac{\alpha_s}{2} \|\boldsymbol{\theta}_1 - \boldsymbol{\theta}_2\|_2^2.$$

**Definition 2** (RSS Property). *A differentiable function $f : \mathbb{R}^p \to \mathbb{R}$ is said to satisfy restricted strong smoothness (RSS) at sparsity level $s = s_1 + s_2$ with strong convexity constraint $L_s$ if the following holds for all $\boldsymbol{\theta}_1, \boldsymbol{\theta}_2$ s.t. $\|\boldsymbol{\theta}_1\|_0 \leq s_1$ and $\|\boldsymbol{\theta}_2\|_0 \leq s_2$:*

$$f(\boldsymbol{\theta}_1) - f(\boldsymbol{\theta}_2) \leq \langle \boldsymbol{\theta}_1 - \boldsymbol{\theta}_2, \nabla_{\boldsymbol{\theta}} f(\boldsymbol{\theta}_2) \rangle + \frac{L_s}{2} \|\boldsymbol{\theta}_1 - \boldsymbol{\theta}_2\|_2^2.$$

**Low-rank Matrix Regression.** Low-rank matrix regression is similar to sparse estimation as presented above except that each data point is now a matrix i.e. $X_i \in \mathbb{R}^{p_1 \times p_2}$, the goal being to estimate a low-rank matrix $W \in \mathbb{R}^{p_1 \times p_2}$ that minimizes the empirical loss function on the given data.

$$W^* = \arg \min_{W, rank(W) \leq r} f(W). \tag{2}$$

For this problem the RSC and RSS properties for $f$ are defined similarly as in Definition 1, 2 except that the $L_0$ norm is replaced by the rank function.

## 3 Iterative Hard-thresholding Method

In this section we study the popular projected gradient descent (a.k.a iterative hard thresholding) method for the case of the feasible set being the set of sparse vectors (see Algorithm 1 for pseudocode). The projection operator $P_s(\boldsymbol{z})$, can be implemented efficiently in this case by projecting $\boldsymbol{z}$ onto the set of $s$-sparse vectors by selecting the $s$ largest elements (in magnitude) of $\boldsymbol{z}$. The standard projection property implies that $\|P_s(\boldsymbol{z}) - \boldsymbol{z}\|_2^2 \leq \|\boldsymbol{\theta}' - \boldsymbol{z}\|_2^2$ for all $\|\boldsymbol{\theta}'\|_0 \leq s$. However, it turns out that we can prove a significantly stronger property of hard thresholding for the case when $\|\boldsymbol{\theta}'\|_0 \leq s^*$ and $s^* \ll s$. This property is key to analysing IHT and is formalized below.

**Lemma 1.** *For any index set $I$, any $\boldsymbol{z} \in \mathbb{R}^I$, let $\boldsymbol{\theta} = P_s(\boldsymbol{z})$. Then for any $\boldsymbol{\theta}^* \in \mathbb{R}^I$ such that $\|\boldsymbol{\theta}^*\|_0 \leq s^*$, we have*

$$\|\boldsymbol{\theta} - \boldsymbol{z}\|_2^2 \leq \frac{|I| - s}{|I| - s^*} \|\boldsymbol{\theta}^* - \boldsymbol{z}\|_2^2.$$

See Appendix A for a detailed proof.

Our analysis combines the above observation with the RSC/RSS properties of $f$ to provide geometric convergence rates for the IHT procedure below.

---

**Algorithm 1** Iterative Hard-thresholding

---
1: **Input**: Function $f$ with gradient oracle, sparsity level $s$, step-size $\eta$
2: $\boldsymbol{\theta}^1 = \mathbf{0}$, $t = 1$
3: **while** *not converged* **do**
4: $\quad \boldsymbol{\theta}^{t+1} = P_s(\boldsymbol{\theta}^t - \eta\nabla_{\boldsymbol{\theta}} f(\boldsymbol{\theta}^t))$, $t = t+1$
5: **end while**
6: **Output**: $\boldsymbol{\theta}^t$

---

**Theorem 1.** *Let $f$ have RSC and RSS parameters given by $L_{2s+s^*}(f) = L$ and $\alpha_{2s+s^*}(f) = \alpha$ respectively. Let Algorithm 1 be invoked with $f$, $s \geq 32\left(\frac{L}{\alpha}\right)^2 s^*$ and $\eta = \frac{2}{3L}$. Also let $\boldsymbol{\theta}^* = \arg\min_{\boldsymbol{\theta}, \|\boldsymbol{\theta}\|_0 \leq s^*} f(\boldsymbol{\theta})$. Then, the $\tau$-th iterate of Algorithm 1, for $\tau = O(\frac{L}{\alpha} \cdot \log(\frac{f(\boldsymbol{\theta}^0)}{\epsilon}))$ satisfies:*
$$f(\boldsymbol{\theta}^\tau) - f(\boldsymbol{\theta}^*) \leq \epsilon.$$

*Proof.* (Sketch) Let $S^t = supp(\boldsymbol{\theta}^t)$, $S^* = supp(\boldsymbol{\theta}^*)$, $S^{t+1} = supp(\boldsymbol{\theta}^{t+1})$ and $I^t = S^* \cup S^t \cup S^{t+1}$. Using the RSS property and the fact that $supp(\boldsymbol{\theta}^t) \subseteq I^t$ and $supp(\boldsymbol{\theta}^{t+1}) \subseteq I^t$, we have:

$$f(\boldsymbol{\theta}^{t+1}) - f(\boldsymbol{\theta}^t) \leq \langle \boldsymbol{\theta}^{t+1} - \boldsymbol{\theta}^t, \boldsymbol{g}^t \rangle + \frac{L}{2}\|\boldsymbol{\theta}^{t+1} - \boldsymbol{\theta}^t\|_2^2,$$

$$= \frac{L}{2}\|\boldsymbol{\theta}_{I^t}^{t+1} - \boldsymbol{\theta}_{I^t}^t + \frac{2}{3L} \cdot \boldsymbol{g}_{I^t}^t\|_2^2 - \frac{1}{2L}\|\boldsymbol{g}_{I^t}^t\|_2^2,$$

$$\overset{\zeta_1}{\leq} \frac{L}{2} \cdot \frac{|I^t| - s}{|I^t| - s^*} \cdot \|\boldsymbol{\theta}_{I^t}^* - \boldsymbol{\theta}_{I^t}^t + \frac{1}{L} \cdot \boldsymbol{g}_{I^t}^t\|_2^2 - \frac{1}{2L}(\|\boldsymbol{g}_{I^t \setminus (S^t \cup S^*)}^t\|_2^2 + \|\boldsymbol{g}_{S^t \cup S^*}^t\|_2^2), \tag{3}$$

where $\zeta_1$ follows from an application of Lemma 1 with $I = I^t$ and the Pythagoras theorem. The above equation has three critical terms. The first term can be bounded using the RSS condition. Using $f(\boldsymbol{\theta}^t) - f(\boldsymbol{\theta}^*) \leq \langle \boldsymbol{g}_{S^t \cup S^*}^t, \boldsymbol{\theta}^t - \boldsymbol{\theta}^* \rangle - \frac{\alpha}{2}\|\boldsymbol{\theta}^t - \boldsymbol{\theta}^*\|_2^2 \leq \frac{1}{2\alpha}\|\boldsymbol{g}_{S^t \cup S^*}^t\|_2^2$ bounds the third term in (3). The second term is more interesting as in general elements of $\boldsymbol{g}_{S^*}^t$ can be arbitrarily small. However, elements of $\boldsymbol{g}_{I^t \setminus (S^t \cup S^*)}^t$ should be at least as large as $\boldsymbol{g}_{S^* \setminus S^{t+1}}^t$ as they are selected by hard-thresholding. Combining this insight with bounds for $\boldsymbol{g}_{S^* \setminus S^{t+1}}^t$ and with (3), we obtain the theorem. See Appendix A for a detailed proof. $\square$

### 3.1 Low-rank Matrix Regression

We now generalize our previous analysis to a projected gradient descent (PGD) method for low-rank matrix regression. Formally, we study the following problem:
$$\min_W f(W), \ s.t., \ rank(W) \leq s. \tag{4}$$
The hard-thresholding projection step for low-rank matrices can be solved using SVD i.e.
$$PM_s(W) = U_s\Sigma_s V_s^T,$$
where $W = U\Sigma V^T$ is the singular value decomposition of $W$. $U_s, V_s$ are the top-$s$ singular vectors (left and right, respectively) of $W$ and $\Sigma_s$ is the diagonal matrix of the top-$s$ singular values of $W$. To proceed, we first note a property of the above projection similar to Lemma 1.

**Lemma 2.** *Let $W \in \mathbb{R}^{p_1 \times p_2}$ be a rank-$|I^t|$ matrix and let $p_1 \geq p_2$. Then for any rank-$s^*$ matrix $W^* \in \mathbb{R}^{p_1 \times p_2}$ we have*
$$\|PM_s(W) - W\|_F^2 \leq \frac{|I^t| - s}{|I^t| - s^*}\|W^* - W\|_F^2. \tag{5}$$

*Proof.* Let $W = U\Sigma V^T$ be the singular value decomposition of $W$. Now, $\|PM_s(W) - W\|_F^2 = \sum_{i=s+1}^{|I^t|} \sigma_i^2 = \|P_s(diag(\Sigma)) - diag(\Sigma)\|_2^2$, where $\sigma_1 \geq \cdots \geq \sigma_{|I^t|} \geq 0$ are the singular values of $W$. Using Lemma 1, we get:
$$\|PM_s(W) - W\|_F^2 \leq \frac{|I^t| - s}{|I^t| - s^*}\|\Sigma^* - diag(\Sigma)\|_2^2 \leq \frac{|I^t| - s}{|I^t| - s^*}\|W^* - W\|_F^2, \tag{6}$$
where the last step uses the von Neumann's trace inequality $(Tr(A \cdot B) \leq \sum_i \sigma_i(A)\sigma_i(B))$. $\square$

The following result for low-rank matrix regression immediately follows from Lemma 4.

**Theorem 2.** *Let $f$ have RSC and RSS parameters given by $L_{2s+s^*}(f) = L$ and $\alpha_{2s+s^*}(f) = \alpha$. Replace the projection operator $P_s$ in Algorithm 1 with its matrix counterpart $PM_s$ as defined in* (5). *Suppose we invoke it with $f, s \geq 32\left(\frac{L}{\alpha}\right)^2 s^*, \eta = \frac{2}{3L}$. Also let $W^* = \arg\min_{W, rank(W) \leq s^*} f(W)$. Then the $\tau$-th iterate of Algorithm 1, for $\tau = O(\frac{L}{\alpha} \cdot \log(\frac{f(W^0)}{\epsilon}))$ satisfies:*

$$f(W^\tau) - f(W^*) \leq \epsilon.$$

*Proof.* A proof progression similar to that of Theorem 1 suffices. The only changes that need to be made are: firstly Lemma 2 has to be invoked in place of Lemma 1. Secondly, in place of considering vectors restricted to a subset of coordinates viz. $\boldsymbol{\theta}_S, \boldsymbol{g}_I^t$, we would need to consider matrices restricted to subspaces i.e. $W_S = U_S U_S^T W$ where $U_S$ is a set of singular vectors spanning the range-space of $S$. □

## 4 High Dimensional Statistical Estimation

This section elaborates on how the results of the previous section can be used to give guarantees for IHT-style techniques in a variety of statistical estimation problems. We will first present a generic convergence result and then specialize it to various settings. Suppose we have a sample of data points $Z_{1:n}$ and a loss function $\mathcal{L}(\boldsymbol{\theta}; Z_{1:n})$ that depends on a parameter $\boldsymbol{\theta}$ and the sample. Then we can show the following result. (See Appendix B for a proof.)

**Theorem 3.** *Let $\bar{\boldsymbol{\theta}}$ be any $s^*$-sparse vector. Suppose $\mathcal{L}(\boldsymbol{\theta}; Z_{1:n})$ is differentiable and satisfies RSC and RSS at sparsity level $s + s^*$ with parameters $\alpha_{s+s^*}$ and $L_{s+s^*}$ respectively, for $s \geq 32\left(\frac{L_{2s+s^*}}{\alpha_{2s+s^*}}\right)^2 s^*$. Let $\boldsymbol{\theta}^\tau$ be the $\tau$-th iterate of Algorithm 1 for $\tau$ chosen as in Theorem 1 and $\varepsilon$ be the function value error incurred by Algorithm 1. Then we have*

$$\|\bar{\boldsymbol{\theta}} - \boldsymbol{\theta}^\tau\|_2 \leq \frac{2\sqrt{s+s^*}\|\nabla\mathcal{L}(\bar{\boldsymbol{\theta}}; Z_{1:n})\|_\infty}{\alpha_{s+s^*}} + \sqrt{\frac{2\epsilon}{\alpha_{s+s^*}}}.$$

Note that the result does *not* require the loss function to be convex. This fact will be crucially used later. We now apply the above result to several statistical estimation scenarios.

**Sparse Linear Regression.** Here $Z_i = (X_i, Y_i) \in \mathbb{R}^p \times \mathbb{R}$ and $Y_i = \langle \bar{\boldsymbol{\theta}}, X_i \rangle + \xi_i$ where $\xi_i \sim \mathcal{N}(0, \sigma^2)$ is label noise. The empirical loss is the usual least squares loss i.e. $\mathcal{L}(\boldsymbol{\theta}; Z_{1:n}) = \frac{1}{n}\|Y - X\boldsymbol{\theta}\|_2^2$. Suppose $X_{1:n}$ are drawn i.i.d. from a sub-Gaussian distribution with covariance $\Sigma$ with $\Sigma_{jj} \leq 1$ for all $j$. Then [22, Lemma 6] immediately implies that RSC and RSS at sparsity level $k$ hold, with probability at least $1 - e^{-c_0 n}$, with $\alpha_k = \frac{1}{2}\sigma_{\min}(\Sigma) - c_1\frac{k\log p}{n}$ and $L_k = 2\sigma_{\max}(\Sigma) + c_1\frac{k\log p}{n}$ ($c_0, c_1$ are universal constants). So we can set $k = 2s + s^*$ and if $n > 4c_1 k\log p/\sigma_{\min}(\Sigma)$ then we have $\alpha_k \geq \frac{1}{4}\sigma_{\min}(\Sigma)$ and $L_k \leq 2.25\sigma_{\max}(\Sigma)$ which means that $L_k/9\alpha_k \leq \kappa(\Sigma) := \sigma_{\max}(\Sigma)/\sigma_{\min}(\Sigma)$. Thus it is enough to choose $s = 2592\kappa(\Sigma)^2 s^*$ and apply Theorem 3. Note that $\|\nabla\mathcal{L}(\bar{\boldsymbol{\theta}}; Z_{1:n})\|_\infty = \|X^T\xi/n\|_\infty \leq 2\sigma\sqrt{\frac{\log p}{n}}$ with probability at least $1 - c_2 p^{-c_3}$ ($c_2, c_3$ are universal constants). Putting everything together, we have the following bound with high probability:

$$\|\bar{\boldsymbol{\theta}} - \boldsymbol{\theta}^\tau\|_2 \leq 145\frac{\kappa(\Sigma)}{\sigma_{\min}(\Sigma)}\sigma\sqrt{\frac{s^*\log p}{n}} + 2\sqrt{\frac{\epsilon}{\sigma_{\min}(\Sigma)}},$$

where $\epsilon$ is the function value error incurred by Algorithm 1.

**Noisy and Missing Data.** We now look at cases with feature noise as well. More specifically, assume that we only have access to $\tilde{X}_i$'s that are corrupted versions of $X_i$'s. Two models of noise are popular in literature [21]: a) (*additive noise*) $\tilde{X}_i = X_i + W_i$ where $W_i \sim \mathcal{N}(\mathbf{0}, \Sigma_W)$, and b) (*missing data*) $\tilde{X}$ is an $\mathbb{R} \cup \{\star\}$-valued matrix obtained by independently, with probability $\nu \in [0, 1)$, replacing each entry in $X$ with $\star$. For the case of additive noise (missing data can be handled similarly), $Z_i = (\tilde{X}_i, Y_i)$ and $\mathcal{L}(\boldsymbol{\theta}; Z_{1:n}) = \frac{1}{2}\boldsymbol{\theta}^T\hat{\Gamma}\boldsymbol{\theta} - \hat{\gamma}^T\boldsymbol{\theta}$ where $\hat{\Gamma} = \tilde{X}^T\tilde{X}/n - \Sigma_W$ and $\hat{\gamma} = \tilde{X}^T Y/n$ are

---

**Algorithm 2** Two-stage Hard-thresholding

---
1: **Input**: function $f$ with gradient oracle, sparsity level $s$, sparsity expansion level $\ell$
2: $\boldsymbol{\theta}^1 = 0$, $t = 1$
3: **while** *not converged* **do**
4: $\quad \boldsymbol{g}^t = \nabla_{\boldsymbol{\theta}} f(\boldsymbol{\theta}^t)$, $S^t = supp(\boldsymbol{\theta}^t)$
5: $\quad Z^t = S^t \cup$ (largest $\ell$ elements of $|\boldsymbol{g}^t_{\overline{S^t}}|$)
6: $\quad \boldsymbol{\beta}^t = \arg\min_{\boldsymbol{\beta}, supp(\beta) \subseteq Z^t} f(\boldsymbol{\beta})$            // fully corrective step
7: $\quad \widetilde{\boldsymbol{\theta}}^t = P_s(\boldsymbol{\beta}^t)$
8: $\quad \boldsymbol{\theta}^{t+1} = \arg\min_{\boldsymbol{\theta}, supp(\boldsymbol{\theta}) \subseteq supp(\widetilde{\boldsymbol{\theta}}^t)} f(\boldsymbol{\theta})$, $t = t + 1$      // fully corrective step
9: **end while**
10: **Output**: $\boldsymbol{\theta}^t$

---

unbiased estimators of $\Sigma$ and $\Sigma^T \bar{\boldsymbol{\theta}}$ respectively. [21, Appendix A, Lemma 1] implies that RSC, RSS at sparsity level $k$ hold, with failure probability exponentially small in $n$, with $\alpha_k = \frac{1}{2}\sigma_{\min}(\Sigma) - k\tau(p)/n$ and $L_k = 1.5\sigma_{\max}(\Sigma) + k\tau(p)/n$ for $\tau(p) = c_0\sigma_{\min}(\Sigma)\max(\frac{(\|\Sigma\|_{\mathrm{op}}^2 + \|\Sigma_W\|_{\mathrm{op}}^2)^2}{\sigma_{\min}^2(\Sigma)}, 1)\log p$. Thus for $k = 2s + s^*$ and $n \geq 4k\tau(p)/\sigma_{\min}(\Sigma)$ we have $L_k/\alpha_k \leq 7\kappa(\Sigma)$. Note that $\mathcal{L}(\cdot; Z_{1:n})$ is *non-convex* but we can still apply Theorem 3 with $s = 1568\kappa(\Sigma)^2 s^*$ because RSC, RSS hold. Using the high probability upper bound (see [21, Appendix A, Lemma 2]) $\|\nabla\mathcal{L}(\bar{\boldsymbol{\theta}}; Z_{1:n})\|_\infty \leq c_1\tilde{\sigma}\|\bar{\boldsymbol{\theta}}\|_2\sqrt{\log p/n}$ gives us the following

$$\|\bar{\boldsymbol{\theta}} - \boldsymbol{\theta}^\tau\|_2 \leq c_2\frac{\kappa(\Sigma)}{\sigma_{\min}(\Sigma)}\tilde{\sigma}\|\bar{\boldsymbol{\theta}}\|_2\sqrt{\frac{s^*\log p}{n}} + 2\sqrt{\frac{\epsilon}{\sigma_{\min}(\Sigma)}}$$

where $\tilde{\sigma} = \sqrt{\|\Sigma_W\|_{\mathrm{op}}^2 + \|\Sigma\|_{\mathrm{op}}^2}(\|\Sigma_W\|_{\mathrm{op}} + \sigma)$ and $\epsilon$ is the function value error in Algorithm 1.

## 5 Fully-corrective Methods

In this section, we study a variety of "fully-corrective" methods. These methods keep the optimization objective fully minimized over the support of the current iterate. To this end, we first prove a fundamental theorem for fully-corrective methods that formalizes the intuition that for such methods, a large function value should imply a large gradient at any sparse $\boldsymbol{\theta}$ as well. This result is similar to Lemma 1 of [17] but holds under RSC/RSS conditions (rather than the RIP condition as in [17]), as well as for the general loss functions. See Appendix C for a detailed proof.

**Lemma 3.** *Consider a function $f$ with RSC parameter given by $L_{2s+s^*}(f) = L$ and RSS parameter given by $\alpha_{2s+s^*}(f) = \alpha$. Let $\boldsymbol{\theta}^* = \arg\min_{\boldsymbol{\theta}, \|\boldsymbol{\theta}\|_0 \leq s^*} f(\boldsymbol{\theta})$ with $S^* = supp(\boldsymbol{\theta}^*)$. Let $S^t \subseteq [p]$ be any subset of co-ordinates s.t. $|S^t| \leq s$. Let $\boldsymbol{\theta}^t = \arg\min_{\boldsymbol{\theta}, supp(\boldsymbol{\theta}) \subseteq S^t} f(\boldsymbol{\theta})$. Then, we have:*

$$2\alpha(f(\boldsymbol{\theta}^t) - f(\boldsymbol{\theta}^*)) \leq \|\boldsymbol{g}^t_{S^t \cup S^*}\|_2^2 - \alpha^2\|\boldsymbol{\theta}^t_{S^t \setminus S^*}\|_2^2$$

**Two-stage Methods.** We will, for now, concentrate on a family of two-stage fully corrective methods that contains popular compressive sensing algorithms like CoSaMP and Subspace Pursuit (see Algorithm 2 for pseudocode). These algorithms have thus far been analyzed only under RIP conditions for the least squares objective. Using our analysis framework developed in the previous sections, we present a generic RSC/RSS-based analysis for general two-stage methods for arbitrary loss functions. Our analysis shall use the following key observation that the the hard thresholding step in two stage methods does not increase the objective function a lot.

We defer the analysis of *partial hard thresholding* methods to a later version of the paper. This family includes the OMPR($\ell$) method [17], which is known to provide the best known RIP guarantees in the compressive sensing setting. Using our proof techniques, we can show that this method offers geometric convergence rates in the statistical setting as well.

**Lemma 4.** *Let $Z_t \subseteq [n]$ and $|Z_t| \leq q$. Let $\boldsymbol{\beta}^t = \arg\min_{\boldsymbol{\beta}, supp(\beta) \subseteq Z_t} f(\boldsymbol{\beta})$ and $\widehat{\boldsymbol{\theta}}^t = P_q(\boldsymbol{\beta}^t)$. Then, the following holds:*

$$f(\widehat{\boldsymbol{\theta}}^t) - f(\boldsymbol{\beta}^t) \leq \frac{L}{\alpha} \cdot \frac{\ell}{s + \ell - s^*} \cdot (f(\boldsymbol{\beta}^t) - f(\boldsymbol{\theta}^*)).$$

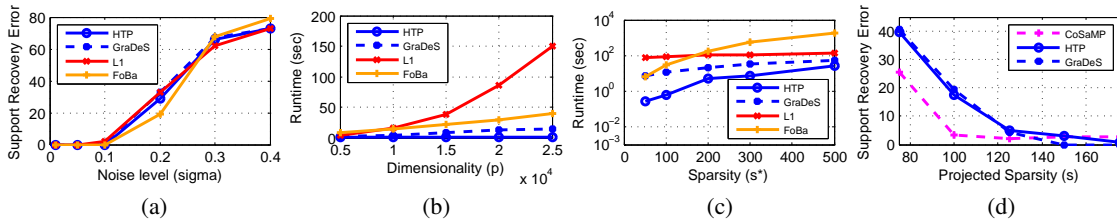

Figure 1: A comparison of hard thresholding techniques (HTP) and projected gradient methods (GraDeS) with L1 and greedy methods (FoBa) on sparse noisy linear regression tasks. 1(a) gives the number of undiscovered elements from $supp(\boldsymbol{\theta}^*)$ as label noise levels are increased. 1(b) shows the variation in running times with increasing dimensionality $p$. 1(c) gives the variation in running times (in logscale) when the true sparsity level $s^*$ is increased keeping $p$ fixed. HTP and GraDeS are clearly much more scalable than L1 and FoBa. 1(d) shows the recovery properties of different IHT methods under large condition number ($\kappa = 50$) setting as the size of projected set is increased.

*Proof.* Let $\boldsymbol{v}^t = \nabla_{\boldsymbol{\theta}} f(\boldsymbol{\beta}^t)$. Then, using the RSS property we get:

$$f(\widehat{\boldsymbol{\theta}}^t) - f(\boldsymbol{\beta}^t) \leq \langle \widehat{\boldsymbol{\theta}}^t - \boldsymbol{\beta}^t, \boldsymbol{v}^t \rangle + \frac{L}{2}\|\widehat{\boldsymbol{\theta}}^t - \boldsymbol{\beta}^t\|_2^2 \overset{\zeta_1}{=} \frac{L}{2}\|\widehat{\boldsymbol{\theta}}^t - \boldsymbol{\beta}^t\|_2^2 \overset{\zeta_2}{\leq} \frac{L}{2}\frac{|\ell|}{|s + \ell - s^*|}\|w - \boldsymbol{\beta}^t\|_2^2, \quad (7)$$

where $w$ is any vector such that $w_{\overline{Z_t}} = 0$ and $\|w\|_0 \leq s^*$. $\zeta_1$ follows by observing $\boldsymbol{v}_{Z_t}^t = 0$ and by noting that $supp(\widehat{\boldsymbol{\theta}}^t) \subseteq Z_t$. $\zeta_2$ follows by Lemma 1 and the fact that $\|w\|_0 \leq s^*$. Now, using the RSC property and the fact that $\nabla_{\boldsymbol{\theta}} f(\boldsymbol{\beta}^t) = 0$, we have:

$$\frac{\alpha}{2}\|w - \boldsymbol{\beta}^t\|_2^2 \leq f(\boldsymbol{\beta}^t) - f(w) \leq f(\boldsymbol{\beta}^t) - f(\boldsymbol{\theta}^*). \quad (8)$$

The result now follows by combining (7) and (8). $\qquad\square$

**Theorem 4.** *Let $f$ have RSC and RSS parameters given by $\alpha_{2s+s^*}(f) = \alpha$ and $L_{2s+\ell}(f) = L$ resp. Call Algorithm 2 with $f$, $\ell \geq s^*$ and $s \geq 4\frac{L^2}{\alpha^2}\ell + s^* - \ell \geq 4\frac{L^2}{\alpha^2}s^*$. Also let $\boldsymbol{\theta}^* = \arg\min_{\boldsymbol{\theta}, \|\boldsymbol{\theta}\|_0 \leq s^*} f(\boldsymbol{\theta})$. Then, the $\tau$-th iterate of Algorithm 2, for $\tau = O(\frac{L}{\alpha} \cdot \log(\frac{f(\boldsymbol{\theta}^0)}{\epsilon}))$ satisfies:*

$$f(\boldsymbol{\theta}^\tau) - f(\boldsymbol{\theta}^*) \leq \epsilon.$$

See Appendix C for a detailed proof.

# 6 Experiments

We conducted simulations on high dimensional sparse linear regression problems to verify our predictions. Our experiments demonstrate that hard thresholding and projected gradient techniques can not only offer recovery in stochastic setting, but offer much more scalable routines for the same.

**Data**: Our problem setting is identical to the one described in the previous section. We fixed a parameter vector $\bar{\boldsymbol{\theta}}$ by choosing $s^*$ random coordinates and setting them randomly to $\pm 1$ values. Data samples were generated as $Z_i = (X_i, Y_i)$ where $X_i \sim \mathcal{N}(0, I_p)$ and $Y_i = \langle \bar{\boldsymbol{\theta}}, X_i \rangle + \xi_i$ where $\xi_i \sim \mathcal{N}(0, \sigma^2)$. We studied the effect of varying dimensionality $p$, sparsity $s^*$, sample size $n$ and label noise level $\sigma$ on the recovery properties of the various algorithms as well as their run times. We chose baseline values of $p = 20000, s^* = 100, \sigma = 0.1, n = f_o \cdot s^* \log p$ where $f_o$ is the oversampling factor with default value $f_o = 2$. Keeping all other quantities fixed, we varied one of the quantities and generated independent data samples for the experiments.

**Algorithms**: We studied a variety of hard-thresholding style algorithms including HTP [14], GraDeS [13] (or IHT [12]), CoSaMP [15], OMPR [17] and SP [16]. We compared them with a standard implementation of the L1 projected scaled sub-gradient technique [23] for the lasso problem and a greedy method FoBa [24] for the same.

**Evaluation Metrics**: For the baseline noise level $\sigma = 0.1$, we found that all the algorithms were able to recover the support set within an error of $2\%$. Consequently, our focus shifted to running times for these experiments. In the experiments where noise levels were varied, we recorded, for each method, the number of undiscovered support set elements.

**Results**: Figure1 describes the results of our experiments in graphical form. For sake of clarity we included only HTP, GraDeS, L1 and FoBa results in these graphs. Graphs for the other algorithms CoSaMP, SP and OMPR can be seen in the supplementary material. The graphs indicate that whereas hard thresholding techniques are equally effective as L1 and greedy techniques for recovery in noisy settings, as indicated by Figure1(a), the former can be much more efficient and scalable than the latter. For instance, as Figure1(b), for the base level of $p = 20000$, HTP was $150\times$ faster than the L1 method. For higher values of $p$, the runtime gap widened to more than $350\times$. We also note that in both these cases, HTP actually offered exact support recovery whereas L1 was unable to recover 2 and 4 support elements respectively.

Although FoBa was faster than L1 on Figure1(b) experiments, it was still slower than HTP by $50\times$ and $90\times$ for $p = 20000$ and $25000$ respectively. Moreover, due to its greedy and incremental nature, FoBa was found to suffer badly in settings with larger true sparsity levels. As Figure 1(c) indicates, for even moderate sparsity levels of $s^* = 300$ and $500$, FoBa is $60 - 75\times$ slower than HTP. As mentioned before, the reason for this slowdown is the greedy approach followed by FoBa: whereas HTP took less than 5 iterations to converge for these two problems, FoBa spend 300 and 500 iterations respectively. GraDeS was found to offer much lesser run times in comparison being slower than HTP by $30 - 40\times$ for larger values of $p$ and $2 - 5\times$ slower for larger values of $s^*$.

**Experiments on badly conditioned problems.** We also ran experiments to verify the performance of IHT algorithms in high condition number setting. Values of $p, s^*$ and $\sigma$ were kept at baseline levels. After selecting the optimal parameter vector $\bar{\theta}$, we selected $s^*/2$ random coordinates from its support and $s^*/2$ random coordinates outside its support and constructed a covariance matrix with heavy correlations between these chosen coordinates. The condition number of the resulting matrix was close to $50$. Samples were drawn from this distribution and the recovery properties of the different IHT-style algorithms was observed as the projected sparsity levels $s$ were increased. Our results (see Figure 1(d)) corroborate our theoretical observation that these algorithms show a remarkable improvement in recovery properties for ill-conditioned problems with an enlarged projection size.

# 7   Discussion and Conclusions

In our work we studied iterative hard thresholding algorithms and showed that these techniques can offer global convergence guarantees for arbitrary, possibly non-convex, differentiable objective functions, which nevertheless satisfy Restricted Strong Convexity/Smoothness (RSC/RSM) conditions. Our results apply to a large family of algorithms that includes existing algorithms such as IHT, GraDeS, CoSaMP, SP and OMPR. Previously the analyses of these algorithms required stringent RIP conditions that did not allow the (restricted) condition number to be larger than universal constants specific to these algorithms.

Our basic insight was to relax this stringent requirement by running these iterative algorithms with an enlarged support size. We showed that guarantees for high-dimensional M-estimation follow seamlessly from our results by invoking results on RSC/RSM conditions that have already been established in the literature for a variety of statistical settings. Our theoretical results put hard thresholding methods on par with those based on convex relaxation or greedy algorithms. Our experimental results demonstrate that hard thresholding methods outperform convex relaxation and greedy methods in terms of running time, sometime by orders of magnitude, all the while offering competitive or better recovery properties.

Our results apply to sparsity and low rank structure, arguably two of the most commonly used structures in high dimensional statistical learning problems. In future work, it would be interesting to generalize our algorithms and their analyses to more general structures. A unified analysis for general structures will probably create interesting connections with existing unified frameworks such as those based on decomposability [5] and atomic norms [25].

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
