[Supplementary Material · rsc_iht-appendix.pdf]

# A  Proofs for Section 3

*Proof of Lemma 1.* Without loss of generality, assume that we have reordered coordinates such that $|\boldsymbol{z}_1| \geq |\boldsymbol{z}_2| \geq \ldots \geq |\boldsymbol{z}_I|$. Since the projection operator $P_s(\cdot)$ operates by selecting the largest elements by magnitude, we have $\boldsymbol{\theta}_1 = \boldsymbol{z}_1, \ldots, \boldsymbol{\theta}_s = \boldsymbol{z}_s$ and $\boldsymbol{\theta}_{s+1} = \boldsymbol{\theta}_{s+2} = \ldots = \boldsymbol{\theta}_{|I|} = 0$.

Also define $\boldsymbol{\theta}^{\boldsymbol{z}} = P_{s^*}(\boldsymbol{z})$. By the above argument, we have $\boldsymbol{\theta}_1^{\boldsymbol{z}} = \boldsymbol{z}_1, \ldots, \boldsymbol{\theta}_{s^*}^{\boldsymbol{z}} = \boldsymbol{z}_{s^*}$ and $\boldsymbol{\theta}_{s^*+1}^{\boldsymbol{z}} = \boldsymbol{\theta}_{s^*+2}^{\boldsymbol{z}} = \ldots = \boldsymbol{\theta}_{|I|}^{\boldsymbol{z}} = 0$. Now we have

$$\frac{\|\boldsymbol{\theta}^{\boldsymbol{z}} - \boldsymbol{z}\|}{|I| - s^*} - \frac{\|\boldsymbol{\theta} - \boldsymbol{z}\|}{|I| - s} = \frac{1}{|I| - s^*} \sum_{i=s^*+1}^{s} \boldsymbol{z}_i^2 + \left(\frac{1}{|I| - s^*} - \frac{1}{|I| - s}\right) \sum_{i=s+1}^{|I|} \boldsymbol{z}_i^2$$

$$\geq \frac{s - s^*}{|I| - s^*} \boldsymbol{z}_s^2 + \frac{s^* - s}{(|I| - s^*)(|I| - s)}(|I| - s)\boldsymbol{z}_{s+1}^2 \geq 0, \qquad (9)$$

since the coordinates of $\boldsymbol{z}$ are arranged in decreasing order of magnitude. Combining the above with the observation that, due to the projection property $\|\boldsymbol{\theta}^* - \boldsymbol{z}\| \geq \|\boldsymbol{\theta}^{\boldsymbol{z}} - \boldsymbol{z}\|$, proves the result. $\qquad\square$

*Proof of Theorem 1.* Recall that $\boldsymbol{\theta}^{t+1} = P_s(\boldsymbol{\theta}^t - \frac{\eta'}{L}\boldsymbol{g}^t)$ where $\eta' = \frac{2}{3} < 1$. Let $S^t = supp(\boldsymbol{\theta}^t)$, $S^* = supp(\boldsymbol{\theta}^*)$, and $S^{t+1} = supp(\boldsymbol{\theta}^{t+1})$. Also, let $I^t = S^* \cup S^t \cup S^{t+1}$.

Now, using the RSS property and the fact that $supp(\boldsymbol{\theta}^t) \subseteq I^t$ and $supp(\boldsymbol{\theta}^{t+1}) \subseteq I^t$, we have:

$$f(\boldsymbol{\theta}^{t+1}) - f(\boldsymbol{\theta}^t) \leq \langle \boldsymbol{\theta}^{t+1} - \boldsymbol{\theta}^t, \boldsymbol{g}^t \rangle + \frac{L}{2}\|\boldsymbol{\theta}^{t+1} - \boldsymbol{\theta}^t\|_2^2,$$

$$= \frac{L}{2}\|\boldsymbol{\theta}_{I^t}^{t+1} - \boldsymbol{\theta}_{I^t}^t + \frac{\eta'}{L} \cdot \boldsymbol{g}_{I^t}^t\|_2^2 - \frac{(\eta')^2}{2L}\|\boldsymbol{g}_{I^t}^t\|_2^2 + (1 - \eta')\langle \boldsymbol{\theta}^{t+1} - \boldsymbol{\theta}^t, \boldsymbol{g}^t \rangle. \quad (10)$$

As $supp(\boldsymbol{\theta}^t) = S^t$, $supp(\boldsymbol{\theta}^{t+1}) = S^{t+1}$ and $S^t \backslash S^{t+1}, S^{t+1}$ are disjoint, we have:

$$\langle \boldsymbol{\theta}^{t+1} - \boldsymbol{\theta}^t, \boldsymbol{g}^t \rangle = -\langle \boldsymbol{\theta}_{S^t \backslash S^{t+1}}^t, \boldsymbol{g}_{S^t \backslash S^{t+1}}^t \rangle + \langle \boldsymbol{\theta}_{S^{t+1}}^{t+1} - \boldsymbol{\theta}_{S^{t+1}}^t, \boldsymbol{g}_{S^{t+1}}^t \rangle,$$

$$\overset{\zeta_1}{=} -\langle \boldsymbol{\theta}_{S^t \backslash S^{t+1}}^t, \boldsymbol{g}_{S^t \backslash S^{t+1}}^t \rangle - \frac{\eta'}{L}\|\boldsymbol{g}_{S^{t+1}}^t\|_2^2,$$

$$\overset{\zeta_2}{\leq} \frac{\eta'}{2L}\|\boldsymbol{g}_{S^{t+1} \backslash S^t}^t\|_2^2 - \frac{\eta'}{2L}\|\boldsymbol{g}_{S^t \backslash S^{t+1}}^t\|_2^2 - \frac{\eta'}{L}\|\boldsymbol{g}_{S^{t+1}}^t\|_2^2,$$

$$\overset{\zeta_3}{=} -\frac{\eta'}{2L}\|\boldsymbol{g}_{S^{t+1} \backslash S^t}^t\|_2^2 - \frac{\eta'}{2L}\|\boldsymbol{g}_{S^t \backslash S^{t+1}}^t\|_2^2 - \frac{\eta'}{L}\|\boldsymbol{g}_{S^t \cap S^{t+1}}^t\|_2^2$$

$$\leq -\frac{\eta'}{2L}\|\boldsymbol{g}_{S^t \cup S^{t+1}}^t\|_2^2, \qquad (11)$$

where the equality $\zeta_1$ follows from the gradient step, i.e., $\boldsymbol{\theta}_{S^{t+1}}^{t+1} = \boldsymbol{\theta}_{S^{t+1}}^t - \frac{\eta'}{L}\boldsymbol{g}_{S^{t+1}}^t$. The inequality $\zeta_2$ follows using the fact that $\boldsymbol{\theta}^{t+1}$ is obtained using hard thresholding and the fact that $|S^t \backslash S^{t+1}| = |S^{t+1} \backslash S^t|$, as follows:

$$\|\boldsymbol{\theta}_{S^t \backslash S^{t+1}}^t - \frac{\eta'}{L}\boldsymbol{g}_{S^t \backslash S^{t+1}}^t\|_2^2 \leq \|\boldsymbol{\theta}_{S^{t+1} \backslash S^t}^{t+1}\|_2^2 = \frac{(\eta')^2}{L^2}\|\boldsymbol{g}_{S^{t+1} \backslash S^t}^t\|_2^2. \qquad (12)$$

The equality $\zeta_3$ follows from $\|\boldsymbol{g}_{S^{t+1}}^t\|_2^2 = \|\boldsymbol{g}_{S^{t+1} \backslash S^t}^t\|_2^2 + \|\boldsymbol{g}_{S^t \cap S^{t+1}}^t\|_2^2$.

Hence, using (10) and (11), we have:

$$f(\boldsymbol{\theta}^{t+1}) - f(\boldsymbol{\theta}^t) \leq \frac{L}{2}\|\boldsymbol{\theta}_{I^t}^{t+1} - \boldsymbol{\theta}_{I^t}^t + \frac{\eta'}{L} \cdot \boldsymbol{g}_{I^t}^t\|_2^2 - \frac{(\eta')^2}{2L}\|\boldsymbol{g}_{I^t}^t\|_2^2 - \frac{\eta'(1 - \eta')}{2L}\|\boldsymbol{g}_{S^t \cup S^{t+1}}^t\|_2^2,$$

$$= \frac{L}{2}\|\boldsymbol{\theta}_{I^t}^{t+1} - \boldsymbol{\theta}_{I^t}^t + \frac{\eta'}{L} \cdot \boldsymbol{g}_{I^t}^t\|_2^2 - \frac{(\eta')^2}{2L}\|\boldsymbol{g}_{I^t \backslash (S^t \cup S^*)}^t\|_2^2 - \frac{(\eta')^2}{2L}\|\boldsymbol{g}_{S^t \cup S^*}^t\|_2^2$$

$$- \frac{\eta'(1 - \eta')}{2L}\|\boldsymbol{g}_{S^t \cup S^{t+1}}^t\|_2^2. \qquad (13)$$

Next, let us try to upper bound the first two terms on the right hand side above. Since $I^t \backslash (S^t \cup S^*) = S^{t+1} \backslash (S^t \cup S^*) \subseteq S^{t+1}$, we have $\boldsymbol{\theta}_{I^t \backslash (S^t \cup S^*)}^{t+1} = \boldsymbol{\theta}_{I^t \backslash (S^t \cup S^*)}^t - \frac{\eta'}{L}\boldsymbol{g}_{I^t \backslash (S^t \cup S^*)}^t$. However, as

$\boldsymbol{\theta}^t_{I^t \setminus S^t} = 0$, we actually have $\boldsymbol{\theta}^{t+1}_{I^t \setminus (S^t \cup S^*)} = -\frac{\eta'}{L} \boldsymbol{g}^t_{I^t \setminus (S^t \cup S^*)}$. Now let us choose a set $R \subseteq S^t \setminus S^{t+1}$ such that $|R| = |S^{t+1} \setminus (S^t \cup S^*)|$. Such a choice is possible since $|S^{t+1} \setminus (S^t \cup S^*)| = |S^t \setminus S^{t+1}| - |(S^{t+1} \cap S^*) \setminus S^t|$ (which itself is a consequence of the fact that $|S^{t+1}| = |S^t|$). Moreover, since $\boldsymbol{\theta}^{t+1}$ is obtained by hard-thresholding $\left(\boldsymbol{\theta}^t - \frac{\eta'}{L} \boldsymbol{g}^t\right)$, for any choice of $R$ made above, we have:

$$\frac{(\eta')^2}{L^2} \|\boldsymbol{g}^t_{I^t \setminus (S^t \cup S^*)}\|_2^2 = \|\boldsymbol{\theta}^{t+1}_{I^t \setminus (S^t \cup S^*)}\|_2^2 \geq \|\boldsymbol{\theta}^t_R - \frac{\eta'}{L} \boldsymbol{g}^t_R\|_2^2. \tag{14}$$

Using above equation, and the fact that $\boldsymbol{\theta}^{t+1}_R = 0$ (since $R \subseteq \overline{S^{t+1}}$), we have:

$$\begin{aligned}
&\frac{L}{2} \|\boldsymbol{\theta}^{t+1}_{I^t} - \boldsymbol{\theta}^t_{I^t} + \frac{\eta'}{L} \cdot \boldsymbol{g}^t_{I^t}\|_2^2 - \frac{(\eta')^2}{2L} \|\boldsymbol{g}^t_{I^t \setminus (S^t \cup S^*)}\|_2^2 \\
&\leq \frac{L}{2} \|\boldsymbol{\theta}^{t+1}_{I^t} - \boldsymbol{\theta}^t_{I^t} + \frac{\eta'}{L} \cdot \boldsymbol{g}^t_{I^t}\|_2^2 - \frac{L}{2} \|\boldsymbol{\theta}^{t+1}_R - \boldsymbol{\theta}^t_R + \frac{\eta'}{L} \boldsymbol{g}^t_R\|_2^2 \\
&= \frac{L}{2} \|\boldsymbol{\theta}^{t+1}_{I^t \setminus R} - \boldsymbol{\theta}^t_{I^t \setminus R} + \frac{\eta'}{L} \cdot \boldsymbol{g}^t_{I^t \setminus R}\|_2^2. 
\end{aligned} \tag{15}$$

We can bound the size of $I^t \setminus R$ as $|I^t \setminus R| \leq |S^{t+1}| + |(S^t \setminus S^{t+1}) \setminus R| + |S^*| \leq s + |(S^{t+1} \cap S^*) \setminus S^t| + s^* \leq s + 2s^*$. Also, since $S^{t+1} \subseteq (I^t \setminus R)$, we have $\boldsymbol{\theta}^{t+1}_{I^t \setminus R} = P_s(\boldsymbol{\theta}^t_{I^t \setminus R} - \frac{\eta'}{L} \boldsymbol{g}^t_{I^t \setminus R})$.

Using the above observation with (15) and Lemma 1, we get:

$$\begin{aligned}
&\frac{L}{2} \|\boldsymbol{\theta}^{t+1}_{I^t} - \boldsymbol{\theta}^t_{I^t} + \frac{\eta'}{L} \cdot \boldsymbol{g}^t_{I^t}\|_2^2 - \frac{(\eta')^2}{2L} \|\boldsymbol{g}^t_{I^t \setminus (S^t \cup S^*)}\|_2^2 \\
&\leq \frac{L}{2} \cdot \frac{|I^t \setminus R| - s}{|I^t \setminus R| - s^*} \|\boldsymbol{\theta}^*_{I^t \setminus R} - \boldsymbol{\theta}^t_{I^t \setminus R} + \frac{\eta'}{L} \cdot \boldsymbol{g}^t_{I^t \setminus R}\|_2^2, \\
&\overset{\zeta_1}{\leq} \frac{L}{2} \cdot \frac{2s^*}{s + s^*} \|\boldsymbol{\theta}^*_{I^t} - \boldsymbol{\theta}^t_{I^t} + \frac{\eta'}{L} \cdot \boldsymbol{g}^t_{I^t}\|_2^2, \\
&= \frac{2s^*}{s + s^*} \cdot \left(\eta' \langle \boldsymbol{\theta}^* - \boldsymbol{\theta}^t, \boldsymbol{g}^t \rangle + \frac{L}{2} \|\boldsymbol{\theta}^* - \boldsymbol{\theta}^t\|_2^2 + \frac{(\eta')^2}{2L} \|\boldsymbol{g}^t_{I^t}\|_2^2\right), \\
&\overset{\zeta_2}{\leq} \frac{2s^*}{s + s^*} \cdot \left(\eta' f(\boldsymbol{\theta}^*) - \eta' f(\boldsymbol{\theta}^t) + \frac{L - \eta' \alpha}{2} \|\boldsymbol{\theta}^* - \boldsymbol{\theta}^t\|_2^2 + \frac{(\eta')^2}{2L} \|\boldsymbol{g}^t_{I^t}\|_2^2\right),
\end{aligned} \tag{16}$$

where the inequality $\zeta_1$ follows by $|I^t \setminus R| \leq s + 2s^*$ as shown earlier and the observation that $\frac{x-a}{x-b}$ is a positive and increasing function on the interval $x \geq a$ if $a \geq b \geq 0$. Note that since we have $S^{t+1} \subseteq (I^t \setminus R)$, we get $|I^t \setminus R| \geq s$. The inequality $\zeta_2$ follows by using RSC.

Using (13), (16), and using $S^{t+1} \setminus (S^t \cup S^*) \subseteq (S^{t+1} \cup S^t)$, we get:

$$\begin{aligned}
f(\boldsymbol{\theta}^{t+1}) - f(\boldsymbol{\theta}^t) \leq{} & \frac{2s^*}{s + s^*} \cdot \left(\eta' f(\boldsymbol{\theta}^*) - \eta' f(\boldsymbol{\theta}^t) + \frac{L - \eta' \alpha}{2} \|\boldsymbol{\theta}^* - \boldsymbol{\theta}^t\|_2^2 + \frac{(\eta')^2}{2L} \|\boldsymbol{g}^t_{I^t}\|_2^2\right) \\
& - \frac{(\eta')^2}{2L} \|\boldsymbol{g}^t_{S^t \cup S^*}\|_2^2 - \frac{\eta'(1 - \eta')}{2L} \|\boldsymbol{g}^t_{S^{t+1} \setminus (S^t \cup S^*)}\|_2^2.
\end{aligned} \tag{17}$$

We now set $\eta' = 2/3$ as per our earlier choice and set $s = 32 \left(\frac{L}{\alpha}\right)^2 s^*$, so that we have $\frac{2s^*}{s + s^*} \leq \frac{\alpha^2}{16L(L - \eta' \alpha)}$. Since $L \geq \alpha$, we also have $\frac{\alpha^2}{16L(L - \eta' \alpha)} \leq \frac{3}{16}$. Using these inequalities, we now rearrange the terms in (17) above.

$$\begin{aligned}
f(\boldsymbol{\theta}^{t+1}) - f(\boldsymbol{\theta}^t) \leq{} & \frac{2s^*}{s + s^*} \cdot \eta' \cdot \left(f(\boldsymbol{\theta}^*) - f(\boldsymbol{\theta}^t)\right) + \frac{\alpha^2}{32L} \|\boldsymbol{\theta}^* - \boldsymbol{\theta}^t\|_2^2 + \frac{1}{24L} \|\boldsymbol{g}^t_{I^t}\|_2^2 \\
& - \frac{2}{9L} \|\boldsymbol{g}^t_{S^t \cup S^*}\|_2^2 - \frac{1}{9L} \|\boldsymbol{g}^t_{S^{t+1} \setminus (S^t \cup S^*)}\|_2^2.
\end{aligned} \tag{18}$$

Splitting $\|\boldsymbol{g}^t_{I^t}\|_2^2 = \|\boldsymbol{g}^t_{S^t \cup S^*}\|_2^2 + \|\boldsymbol{g}^t_{S^{t+1} \setminus (S^t \cup S^*)}\|_2^2$ gives us

$$f(\boldsymbol{\theta}^{t+1}) - f(\boldsymbol{\theta}^t) \leq \frac{2s^*}{s+s^*} \cdot \eta' \cdot \left( f(\boldsymbol{\theta}^*) - f(\boldsymbol{\theta}^t) \right) - \frac{1}{2L} \left( \frac{13}{36} \|\boldsymbol{g}_{S^t \cup S^*}^t\|_2^2 - \frac{\alpha^2}{16} \|\boldsymbol{\theta}^* - \boldsymbol{\theta}^t\|_2^2 \right)$$

$$- \frac{1}{2L} \cdot \left( \frac{4}{9} - \frac{1}{12} \right) \|\boldsymbol{g}_{S^{t+1} \setminus (S^t \cup S^*)}^t\|_2^2,$$

$$\leq \frac{2s^*}{s+s^*} \cdot \eta' \cdot \left( f(\boldsymbol{\theta}^*) - f(\boldsymbol{\theta}^t) \right) - \frac{13}{72L} \left( \|\boldsymbol{g}_{S^t \cup S^*}^t\|_2^2 - \frac{\alpha^2}{4} \|\boldsymbol{\theta}^* - \boldsymbol{\theta}^t\|_2^2 \right)$$

$$\leq \frac{2s^*}{s+s^*} \cdot \eta' \cdot \left( f(\boldsymbol{\theta}^*) - f(\boldsymbol{\theta}^t) \right) - \frac{\alpha}{12L} \left( f(\boldsymbol{\theta}^t) - f(\boldsymbol{\theta}^*) \right), \tag{19}$$

where the last inequality above follows using Lemma 5. The result now follows by observing that $\frac{2s^*}{s+s^*} \geq 0$. □

**Lemma 5.**

$$\left( \|\boldsymbol{g}_{S^t \cup S^*}^t\|_2^2 - \frac{\alpha^2}{4} \|\boldsymbol{\theta}^* - \boldsymbol{\theta}^t\|_2^2 \right) \geq \frac{\alpha}{2} \cdot \left( f(\boldsymbol{\theta}^t) - f(\boldsymbol{\theta}^*) \right).$$

*Proof.* Using the RSC property, we have:

$$f(\boldsymbol{\theta}^t) - f(\boldsymbol{\theta}^*) \leq \langle \boldsymbol{g}^t, \boldsymbol{\theta}^t - \boldsymbol{\theta}^* \rangle - \frac{\alpha}{2} \|\boldsymbol{\theta}^* - \boldsymbol{\theta}^t\|_2^2$$

$$= \langle \boldsymbol{g}_{S^t \cup S^*}^t, \boldsymbol{\theta}_{S^t \cup S^*}^t - \boldsymbol{\theta}_{S^t \cup S^*}^* \rangle - \frac{\alpha}{2} \|\boldsymbol{\theta}^* - \boldsymbol{\theta}^t\|_2^2,$$

$$\leq \|\boldsymbol{g}_{S^t \cup S^*}^t\|_2 \|\boldsymbol{\theta}^t - \boldsymbol{\theta}^*\|_2 - \frac{\alpha}{2} \|\boldsymbol{\theta}^* - \boldsymbol{\theta}^t\|_2^2. \tag{20}$$

Now,

$$\|\boldsymbol{g}_{S^t \cup S^*}^t\|_2^2 - \frac{\alpha^2}{4} \|\boldsymbol{\theta}^* - \boldsymbol{\theta}^t\|_2^2 = \left( \|\boldsymbol{g}_{S^t \cup S^*}^t\|_2 - \frac{\alpha}{2} \|\boldsymbol{\theta}^* - \boldsymbol{\theta}^t\|_2 \right) \left( \|\boldsymbol{g}_{S^t \cup S^*}^t\|_2 + \frac{\alpha}{2} \|\boldsymbol{\theta}^* - \boldsymbol{\theta}^t\|_2 \right),$$

$$\geq \frac{(f(\boldsymbol{\theta}^t) - f(\boldsymbol{\theta}^*))}{\|\boldsymbol{\theta}^t - \boldsymbol{\theta}^*\|_2} \cdot \left( \|\boldsymbol{g}_{S^t \cup S^*}^t\|_2 + \frac{\alpha}{2} \|\boldsymbol{\theta}^* - \boldsymbol{\theta}^t\|_2 \right)$$

$$\geq \frac{\alpha}{2} \cdot \left( f(\boldsymbol{\theta}^t) - f(\boldsymbol{\theta}^*) \right), \tag{21}$$

where the first inequality above follows from (20). □

# B  Proofs for Section 4

*Proof of Theorem 3.* Let $\boldsymbol{\theta}^*$ be the empirical loss minimizer over the set of $s$-sparse vectors. Then invoking Theorem 1 with $f = \mathcal{L}(\cdot; Z_{1:n})$, we get

$$\mathcal{L}(\boldsymbol{\theta}^\tau, Z_{1:n}) - \epsilon \leq \mathcal{L}(\boldsymbol{\theta}^*, Z_{1:n}) \leq \mathcal{L}(\bar{\boldsymbol{\theta}}, Z_{1:n})$$

$$\leq \mathcal{L}(\boldsymbol{\theta}^\tau; Z_{1:n}) + \langle \nabla \mathcal{L}(\bar{\boldsymbol{\theta}}; Z_{1:n}), (\bar{\boldsymbol{\theta}} - \boldsymbol{\theta}^\tau) \rangle - \frac{\alpha_{s+s^*}}{2} \|\bar{\boldsymbol{\theta}} - \boldsymbol{\theta}^\tau\|_2^2$$

where the 2nd inequality is by definition of $\boldsymbol{\theta}^*$ and 3rd is by RSC (since $\boldsymbol{\theta}^*, \boldsymbol{\theta}^\tau$ are $s^*, s$ sparse). Duality gives us the upper bound

$$\langle \nabla \mathcal{L}(\bar{\boldsymbol{\theta}}; Z_{1:n}), (\bar{\boldsymbol{\theta}} - \boldsymbol{\theta}^\tau) \rangle \leq \|\nabla \mathcal{L}(\bar{\boldsymbol{\theta}}; Z_{1:n})\|_\infty \|\bar{\boldsymbol{\theta}} - \boldsymbol{\theta}^\tau\|_1 \leq \sqrt{s+s^*} \|\nabla \mathcal{L}(\bar{\boldsymbol{\theta}}; Z_{1:n})\|_\infty \|\bar{\boldsymbol{\theta}} - \boldsymbol{\theta}^\tau\|_2$$

Combining the last two inequalities and rearranging gives a quadratic inequality in $\|\bar{\boldsymbol{\theta}} - \boldsymbol{\theta}^\tau\|_2$:

$$\frac{\alpha_{s+s^*}}{2} \|\bar{\boldsymbol{\theta}} - \boldsymbol{\theta}^\tau\|_2^2 - \sqrt{s+s^*} \|\nabla \mathcal{L}(\bar{\boldsymbol{\theta}}; Z_{1:n})\|_\infty \|\bar{\boldsymbol{\theta}} - \boldsymbol{\theta}^\tau\|_2 - \epsilon \leq 0$$

that immediately yields the result. □

## C Proofs for Section 5

*Proof of Lemma 3.* We will start by proving a more general result of which the claimed result will be a corollary. More specifically, we shall prove that for any $\gamma \geq \frac{1}{\alpha}$, we have

$$2\gamma(f(\boldsymbol{\theta}^t) - f(\boldsymbol{\theta}^*)) \leq 2\gamma\left(f(\boldsymbol{\theta}^t) - f(\boldsymbol{\theta}^*) + \frac{\alpha}{2} \cdot \left(1 - \frac{1}{\alpha\gamma}\right)\|\boldsymbol{\theta}^t - \boldsymbol{\theta}^*\|_2^2\right) \leq \gamma^2\|\boldsymbol{g}_{S^t \cup S^*}^t\|_2^2 - \|\boldsymbol{\theta}_{S^t \setminus S^*}^t\|_2^2,$$

Setting $\gamma = \frac{1}{\alpha}$ will yield the claimed result. It is easy to see that the following inequality holds trivially since $\gamma \geq \frac{1}{\alpha}$

$$2\gamma(f(\boldsymbol{\theta}^t) - f(\boldsymbol{\theta}^*)) \leq 2\gamma\left(f(\boldsymbol{\theta}^t) - f(\boldsymbol{\theta}^*) + \frac{\alpha}{2} \cdot \left(1 - \frac{1}{\alpha\gamma}\right)\|\boldsymbol{\theta}^t - \boldsymbol{\theta}^*\|_2^2\right).$$

For the second inequality, we first use the RSC condition to obtain:

$$f(\boldsymbol{\theta}^*) - f(\boldsymbol{\theta}^t) \geq \langle \boldsymbol{\theta}^* - \boldsymbol{\theta}^t, \boldsymbol{g}^t \rangle + \frac{\alpha}{2}\|\boldsymbol{\theta}^t - \boldsymbol{\theta}^*\|_2^2.$$

Now let $MD_t = S^* \setminus S^t$ be the set of true support elements missing from $\boldsymbol{\theta}^t$ and $FA_t = S^t \setminus S^*$ be the set of incorrect elements included in the support of $\boldsymbol{\theta}^t$. Since $\boldsymbol{\theta}^t$ is obtained by a "fully corrective" process (recall $\boldsymbol{\theta}^t = \arg\min_{\boldsymbol{\theta}, supp(\boldsymbol{\theta}) \subseteq S^t} f(\boldsymbol{\theta})$), we have $\boldsymbol{g}_{S^t}^t = \boldsymbol{0}$. Thus $\langle \boldsymbol{\theta}^* - \boldsymbol{\theta}^t, \boldsymbol{g}^t \rangle = \langle \boldsymbol{\theta}_{MD_t}^*, \boldsymbol{g}_{MD_t}^t \rangle$.

Putting this into the above expansion gives

$$\langle \boldsymbol{\theta}_{MD_t}^*, \boldsymbol{g}_{MD_t}^t \rangle \leq f(\boldsymbol{\theta}^*) - f(\boldsymbol{\theta}^t) - \frac{\alpha}{2}\|\boldsymbol{\theta}^t - \boldsymbol{\theta}^*\|_2^2 \tag{22}$$

We now present some simple inequalities that will help us get our desired bounds. Firstly, we have

$$\|\boldsymbol{\theta}_{MD_t}^* + \gamma\boldsymbol{g}_{MD_t}^t\|_2^2 = \|\boldsymbol{\theta}_{MD_t}^*\|_2^2 + \gamma^2\|\boldsymbol{g}_{MD_t}^t\|_2^2 + 2\gamma\langle \boldsymbol{\theta}_{MD_t}^*, \boldsymbol{g}_{MD_t}^t \rangle \geq 0, \tag{23}$$

since the first expression is a norm. Next, since $MD_t \cap FA_t = \emptyset$, we have

$$\|\boldsymbol{\theta}^* - \boldsymbol{\theta}^t\|_2^2 \geq \|\boldsymbol{\theta}_{MD_t}^*\|_2^2 + \|\boldsymbol{\theta}_{FA_t}^t\|_2^2. \tag{24}$$

Putting equations 22 and 23, we have:

$$2\gamma\left(f(\boldsymbol{\theta}^t) - f(\boldsymbol{\theta}^*) + \frac{\alpha}{2}\|\boldsymbol{\theta}^t - \boldsymbol{\theta}^*\|_2^2\right) \leq \|\boldsymbol{\theta}_{MD_t}^*\|_2^2 + \gamma^2\|\boldsymbol{g}_{MD_t}^t\|_2^2. \tag{25}$$

Now, using (24), we get:

$$2\gamma\left(f(\boldsymbol{\theta}^t) - f(\boldsymbol{\theta}^*) + \frac{\alpha}{2}\left(1 - \frac{1}{\alpha\gamma}\right)\|\boldsymbol{\theta}^t - \boldsymbol{\theta}^*\|_2^2\right) \leq \gamma^2\|\boldsymbol{g}_{MD_t}^t\|_2^2 - \|\boldsymbol{\theta}_{FA_t}^t\|_2^2$$

We finish off the proof by noticing that since $\boldsymbol{g}_{S^t}^t = \boldsymbol{0}$, we have $\|\boldsymbol{g}_{MD_t}^t\|_2^2 = \|\boldsymbol{g}_{S^t \cup S^*}^t\|_2^2$ $\qquad\square$

*Proof of Theorem 4.* Let $\boldsymbol{z}_{S^t}^t = \boldsymbol{\theta}_{S^t}^t$, $\boldsymbol{z}_{Z^t \setminus S^t}^t = -\frac{1}{L}\boldsymbol{g}_{Z^t \setminus S^t}^t$, and $\boldsymbol{z}_{\overline{Z^t}}^t = 0$.

Then, using the RSS property, we have:

$$f(\boldsymbol{z}^t) - f(\boldsymbol{\theta}^t) \leq \langle \boldsymbol{z}^t - \boldsymbol{\theta}^t, \boldsymbol{g}^t \rangle + \frac{L}{2}\|\boldsymbol{z}^t - \boldsymbol{\theta}^t\|_2^2,$$

$$\overset{\zeta_1}{\leq} -\frac{1}{L}\|\boldsymbol{g}_{Z^t \setminus S^t}^t\|_2^2 + \frac{L}{2}\|\boldsymbol{z}_{Z^t \setminus S^t}^t\|_2^2,$$

$$\overset{\zeta_2}{=} -\frac{1}{2L} \cdot \|\boldsymbol{g}_{Z^t \setminus S^t}^t\|_2^2,$$

$$\overset{\zeta_3}{\leq} -\frac{1}{2L} \cdot \|\boldsymbol{g}_{S^* \setminus S^t}^t\|_2^2,$$

$$\overset{\zeta_4}{\leq} -\frac{\alpha}{L} \cdot \left(f(\boldsymbol{\theta}^t) - f(\boldsymbol{\theta}^*)\right), \tag{26}$$

where $\zeta_1$ follows by observing $\boldsymbol{g}_{S^t}^t = 0$, and $S^t \subseteq Z^t$. $\zeta_2$ follows by $\boldsymbol{z}_{Z^t \setminus S^t}^t = -\frac{1}{L}\boldsymbol{g}_{Z^t \setminus S^t}^t$. $\zeta_3$ follows by $\ell \geq s^*$, and $Z^t \setminus S^t$ are the $\ell$ largest elements of $|\boldsymbol{g}_{Z^t \setminus S^t}^t|$.

Now, using Lemma 4 and (26) along with $f(\boldsymbol{\theta}^{t+1}) \leq f(\widetilde{\boldsymbol{\theta}}^t)$ and $f(\boldsymbol{\beta}^t) \leq f(\boldsymbol{z}^t)$, we have:

$$f(\boldsymbol{\theta}^{t+1}) - f(\boldsymbol{\theta}^*) \leq \left(1 - \frac{\alpha}{L}\right) \cdot \left(1 + \frac{L}{\alpha} \cdot \frac{\ell}{s + \ell - s^*}\right) \cdot \left(f(\boldsymbol{\theta}^t) - f(\boldsymbol{\theta}^*)\right). \tag{27}$$

Theorem now follows by using the above equation with the assumption that $s + \ell - s^* \geq \frac{4L^2 \cdot \ell}{\alpha^2}$. $\qquad\square$

# D Supplementary Experimental Results

Below we present plots that were not included in the main text.

(a)  (b)  (c)

Figure 2: Counterparts of Figure 1 for OMPR, CoSaMP and L1.

(a)  (b)

Figure 3: The effect of increasing sample sizes relative to the base value $s^* \cdot \log p$ on runtime.