[Reviews · NeurIPS 2014]

Submitted by Assigned_Reviewer_6

This paper presents a theoretical framework for analyzing hard thresh-holding methods for high dimensional inference. These methods take the general recipe of (in one interpretation) (1) taking a gradient step and (2) projecting this result onto a space such that the result is sparse (typically, the s largest entries of the result). The framework proceeds in a similar fashion as Negahban et al (2012), with smoothness and restricted strong convexity assumptions placed on the objective. The authors first present some general results, and next applications of the results to various relevant problem areas. A small section on related methods to projected gradient (which add an additional 'corrective' step of maximizing the objective on a restricted set). The paper concludes with some simulation studies comparing some thresholding methods to greedy and L_1 methods, which are popular sparsity approaches.

On quality:

Overall, extremely good. The results here are very interesting, and very general. The methods covered by these results are practical, and the applications covered argue strongly for the relevance of the general framework. I can only make a small criticism that the sub-section on GLMs was left as "should be possible"; my intuition is that this should not be a serious challenge, but might also not enough on its own for another paper.

On clarity:

This paper is quite dense, with a large number of both results and pages of supplemental material. The results are substantial and insightful, but there is a lack of text spent on exposing their meaning, or comparing these results to those obtained for other popular approaches (basically I'm thinking of regularized approaches). There is a little of this sort of discussion in the introduction.

As far as the framework is considered, the authors did well giving insight in the proof sketch of Theorem 1. However, some key details are left to the appendix for Theorem 3. In particular, they claim that the loss need not be convex is not expanded upon. The application to noisy data is a key use case of this property, but that part of the proof is given an implication of another proof in the appendix: is there an insight to be gained from it this that could be stated there? There could be a reference that would give insight into where we have restricted convexity but not convexity.

The experiments mostly showed improvements on run-time versus other common classes of approaches. This sort of analysis seems a little out of place given the area of study of the rest of the paper. While many of the results were given in terms of a iteration of a certain order of the algorithm, these ideas can't exactly be lined up with ideas from regularized methods (I admit to not looking up the FoBa results, which might have a similar quality). I'm not really sure that anything better could be done here anyway.

On originality and significance:

Overall, the paper offers highly original and highly significant results. Some parts of the main approach of the framework here are not entirely original. I do not see this as a major detraction from the ideas: I find it sensible that the notions of smoothness and restricted convexity are fundamental here as well.
Summary: This paper presents an insightful and broad theoretical framework for thresholding methods for sparse estimation. The quality and quantity of the results are high, and though lacking somewhat on exposition and intuition for the results, it is overall a very strong submission.

Submitted by Assigned_Reviewer_29

The paper presents analysis for IHT stylized methods for M-estimation. The paper has some interesting ideas.

It may be helpful if the authors acknowledge existing work like [24] and clearly point out the differences between their work and [24]. Upon reading the paper, it appears as if a main contribution of the paper is in extending the framework beyond the squared error loss to "convex" loss functions.
Another paper dealing with similar algorithms that the authors might wish to acknowledge is
http://arxiv.org/abs/1203.4580.
While I agree that the latter paper demonstrates "algorithmic" results (showing convergence to
stationary points) and the current paper has "more" to offer -- they do not come without assuming much more problem structure!

The experimental section is quite weak, the method does not seem to offer any benefit in statistical properties over competing methods. The only difference is in run-time, which I do not think is a fair comparison, since the algorithms being compared are solving different optimization problems.

In various places (of the paper) the authors have remarked that the "IHT"-stylized methods is a method of choice over convex optimization methods---this, in my opinion, is a fairly strong (and possibly misleading) viewpoint.

Summary: The paper has interesting ideas and should be judged by its theoretical merit. The numerical section is weak.

Submitted by Assigned_Reviewer_44

The paper derives a general bound on Iterative hard threshold (IHT) for optimization of (possibly non-convex) functions f(\theta) subject to sparsity constraints on \theta. The problem arises in sparsity-constrained regression / classification problems as well as low-rank optimization. The bounds provided in the paper indicate a sparsity level s* below which the method will work. The sparsity level is based on the ratio of the restricted strong convexity and restricted strong smoothness constants.

The paper seems clearly written with all the definitions nicely spelled out and the results easy to follow. I didn’t fully check all the proofs, but the math appears correct.

The main benefit of the paper is that it applies to a large class of loss functions with simple testable conditions. In particular, the analysis techniques can be used for GLMs.

The only difficulty I had was that it was not clear the exact value of these bounds over previous bounds based on RIP constants. The paper explains correctly that previous bounds on the success of IHT-type algorithms required that the RIP constant (ie. equivalently the condition number restricted to a sparse set) was bounded by a universal constant. This result allows that condition number to be arbitrarily large. However, the resulting bounds do not seem (at least to me) if one looks at the simple sparse linear regression case, we know that LASSO (and I suppose IHT too) will guarantee recovery with a Gaussian ensemble when s* = O(n/log p). But, the equation in this paper requires that s* = o(n/log p). I think the paper would be improved if, when going through the examples, the bounds could be compared to the existing methods.
Summary: A general and potentially useful bound on sparse recovery that applies to a large class of problems. However, the relation to previous bounds, in the cases where they are available, were not made clear and the current result may be weaker.
Author Feedback
Author rebuttal: We thank all reviewers for their comments. Below are responses to the main comments.

Focus of the paper:
This work removes a significant drawback of current analyses of IHT-style methods (such as IHT, CoSamp etc) which is their inability to guarantee recovery for even simple statistical models unless strong structural assumptions as satisfied. Consider the model y= < w, x > + \eta, where x is drawn from N(0, \Sigma). Existing analyses hold only when the condition number of \Sigma is bounded by a global constant say 10, beyond which these methods are not known to work even with *INFINITELY* many samples (see lines 67-79). Here, we provide a novel analysis that shows that by increasing the sparsity parameter of IHT algorithms, we can, in fact match LASSO's min-max optimal bounds for several statistical models.
Some reviewers have questioned the utility of our analysis of IHT methods when LASSO is already known to have optimal min-max bounds. This is readily clarified through our experiments that show that IHT-style algorithms are much faster than LASSO or greedy methods like FoBA, something that has also been observed by several practitioners.

Assigned_Reviewer_29
Comparison to [24]: [24] requires strict universal bounds on the condition number (RSS/RSC, see Defn 1, 2) which means that even for simple Gaussian ensembles (but nevertheless with a bad condition number), their analyses is silent even with infinitely many samples. See lines 67-79 for a detailed discussion with a precise example.

Experiments: High-dimensional data is commonplace and due to the scale of these problems, fast methods are indeed very important. In fact, fast optimization of LASSO has attracted the community's interests for the last 15 years or so. We thus believe that providing provable and fast algorithms for such high-dimensional estimation problems is indeed important for the community.

Comments on the work in arXiv:1203.4580 [cs.IT]: This paper by Beck and Eldar only provides asymptotic convergence guarantees to *stationary* points. We, on the other hand provide linear-time, exact recovery guarantees in a popular statistical setting.

Assigned_Reviewer_44
As mentioned above, RIP constant-based bounds do not handle even simple non-spherical Gaussian ensembles. See lines 67-79 for an example. In contrast, our analyses provide LASSO style min-max optimal bounds for sparse estimation problems.

O(n/log(p)) vs o(n/log(p)): Thanks for pointing this out, this is just a typo in our bounds. Our bounds in fact match LASSO bounds exactly (upto constants), as we provide guarantees under the same RSC property as LASSO; the conversion from RSC property bound to sample complexity bound for Gaussian ensembles being obtained using a result of Agarwal et al. [1].

Assigned_Reviewer_6
Convexity not required: we will highlight this fact more in our proof as well as proof sketch. The paper does give an example of such a loss function (please see the discussion starting line 287).

Experiments: The goal of experiments it to show that although LASSO (and FoBA) have good statistical properties, in practice, their runtime is poor compared to IHT-style methods. Also, we used a state-of-the-art method for optimizing LASSO to make the comparison fair. Moreover, we observe that for many problems, IHT is an order of magnitude faster than LASSO.